# OPTIMIZING ATTENTION

## ABSTRACT

The attention mechanism is an important part of transformer architectures. It enables the network to compare samples within a sequence. Before the comparison is performed, tokens are multiplied by trainable matrices. These matrices can constitute a significant part of the total number of parameters. Their size creates problems on systems with limited cache in the compute unit, especially if there is limited bandwidth between compute unit and memory. In particular, GPUs on mobile devices suffer from this double bottleneck.

Prior works mitigate this problem for instance by storing low-rank approximations, quantization or minimizing the amount of data that needs to be transferred. In this paper, an alternative to the traditional attention mechanism is proposed which does not require any trainable matrices to perform the attention. The idea rests upon solving optimization problems, whereby memory is substituted for compute. It will be shown however, that the computational demand can be reduced such that auto-differentiation becomes possible. An experimental evaluation shows that the proposed algorithm performs favorable compared with several baselines.

## 1 INTRODUCTION

Transformers Vaswani et al. (2017) have been widely used in various applications, such as natural language processing Devlin et al. (2019) or image recognition Dosovitskiy et al. (2021). More recently, very large transformer models have been successfully applied to auto-regressive language modeling, for instance GPT OpenAI et al. (2023), Llama Touvron et al. (2023), and Gemini Bao et al. (2023). These powerful transformers require a vast amount of memory to run due to their large model sizes. While the parameters can be loaded layer by layer during inference, it still poses a significant challenge when running them on mobile devices, which typically have very limited cache in the neural processing unit – often, a few megabytes only. In addition, the attention operation incurs quadratic complexity w.r.t. number of input tokens, both in terms of memory usage and computation.

Recently, researchers have proposed various techniques to reduce the computation and/or memory costs of attention. Efficient attention methods such as Letourneau et al. (2024) only require linear computation and memory costs while maintaining model accuracy. Other works optimize the implementation of attention. For instance, FlashAttention Dao et al. (2022) uses tiling to decompose the softmax computation and other techniques to reduce the number of memory accesses. However, in these approaches, large matrices mapping the input to query, key, and value tensors prior to attention are still necessary. The severely limited cache at the compute units necessitates that only tiny pieces of these matrices can be kept in cache at the same time, thus token data and matrix pieces need to be retrieved from memory multiple times for each attention operation. This increases the amount of data that needs to be transferred between slow main memory and fast cache via limited bandwidth, which incurs high latency especially on resource-constrained devices.

In this work, we propose a novel, optimization-based approach to perform both self- and cross-attention, in order to eliminate both the heavy weight matrices and the computationally expensive *soft-(arg)max* operation during the attention computation. Instead of remixing the values based on the similarity/attention matrix between queries and keys, we directly solve for the mixing coefficients for the value tensor. Specifically, we impose that the remixed values need to be similar to the queries. We additionally impose a sparsity constraint, which leads to a sparse reconstruction problem.

To reduce the high cost for iteratively solving these problems during any forward pass and to reduce the memory demand to allow for backward passes, we propose to use several approximations to increase efficiency, thereby enabling auto-differentiation with the proposed algorithm. To mitigate the errors induced by the proposed techniques, we also propose to include random tokens in our algorithm, similar to the idea proposed by Darcet et al. (2024), yet we offer a geometric intuition regarding the effect these tokens have in the proposed algorithm. A scalar activation and a light-weight normalization further promote sparsity and reduce overfitting.

A well-known property of the standard attention is that the attention coefficients, namely the *soft-(arg)max* normalized scalar products between queries and keys, are very sparse, and thus focus on a few key/value tokens. For images, it has been shown, e.g., by Dosovitskiy et al. (2021), that high attention weights are concentrated on a few spots in the image from which the keys stem. Our proposed approach achieves similar effects, but achieves this via sparse signal reconstruction. This offers a new interpretation of the mechanism behind attention, as well as a new way to compute attention by solving the sparse signal reconstruction problem. In other words, the proposed algorithm optimizes over the given data to determine where to focus attention, ie where to place high attention scores.

An experimental evaluation shows that the proposed algorithm compares favorable with suitable baselines. We provide qualitative example of semantically meaningful attention maps.

This paper is structured as follows: In Sec. 2, we briefly summarize related works. We give a detailed explanation of the standard attention mechanism in Sec. 3 and present our solution in Sec. 4. The experimental evaluation follows in Sec. 5. The paper concludes with a discussion in Sec. 6.

## 2 RELATED WORK

**Reducing Model Sizes**    Jacob et al. (2018) proposed to reduce model sizes by quantization. It is a technique to reduce the memory requirements by converting the weights from some higher-precision representation, such as 32-bit floating point numbers, to some lower precision, often integers. This process not only reduces the model sizes but may also increase the efficiency if the compute unit can process integer data faster.

Model sizes can also be reduced by so-called low-rank approximation (Guo et al., 2023) that is to approximate matrices by low-rank approximations. Only the rank-reduced variants are being stored. As with quantization, LoRA-approximation can also increase the time data needs to travel through the network. Other techniques to reduce model sizes are pruning (Sun et al., 2024) or distillation (Oquab et al., 2023).

**Computational Complexity of Attention**    Choromanski et al. (2020) used random orthogonal features to estimate the a low-rank approximation of the attention matrix. Child et al. (2019) proposed to use attention masks with pre-defined sparsity patterns to reduce the computational demand of attention. Locality-sensitive hashing is used by Kitaev et al. (2019) in the attention, so that only those key tokens that are closest to each query are being used.

A transformer with linear complexity for computing self-attention was proposed by Wang et al. (2020), albeit only for fixed sequence-lengths. Katharopoulos et al. (2020) proposed an attention mechanism with linear complexity for variable sequence lengths. You et al. (2023) improved this idea by using an approximation to angular similarities as activation.

State-Space-Models (Poli et al., 2023), (Gu & Dao, 2023), (Patro & Agneeswaran, 2024) model the interaction between tokens by first-order differential equations or approximations thereof. For images, Hou et al. (2024) use a convolutional architecture that employs self-attention. To reduce the computational complexity of this operation, the authors propose to approximate the attention matrix by 2D-convolutions. Letourneau et al. (2024) generalize this idea by approximating the self-attention with polynomials. They show that their algorithm generalizes several others, for instance Conv2Former (Hou et al., 2024) or (Gu & Dao, 2023).

FlashAttention by (Dao et al., 2022) was proposed to minimize the number of accesses to the main memory of the GPU since the bandwidth between the compute units to the memory is limited. Follow-up works Dao (2024); Shah et al. (2024) further optimize the idea. Since evaluating the

*soft-(arg)argmax* operator is compute-expensive, Koohpayegani & Pirsiavash (2024) proposed a transformer variant that replaces this operation.

## 3   STANDARD ATTENTION MECHANISM

Attention performs exhaustive pairwise comparisons between the set of query tokens, $\boldsymbol{q}_1, \ldots, \boldsymbol{q}_{N_q}$, with the set of key tokens, $\boldsymbol{k}_1, \ldots, \boldsymbol{k}_{N_k}$. The resulting similarity matrix, i.e., attention matrix, is used to linearly combine a set of value tokens, $\boldsymbol{v}_1, \ldots, \boldsymbol{v}_{N_k}$.

In self-attention, the elements, $\boldsymbol{q}_i, \boldsymbol{k}_j, \boldsymbol{v}_j$, are created by multiplying the input by the weight matrices, $\boldsymbol{W}_q, \boldsymbol{W}_k$, and $\boldsymbol{W}_v$, which correspond to the queries, keys, and values, respectively. The weight matrices have sizes $D \times D_{q,k,v}$. Here, the three numbers $D_{q,k,v}$ indicate the dimensions of the three token sequences $\boldsymbol{q}_i, \boldsymbol{k}_j$, and $\boldsymbol{v}_j$.

In cross-attention, two different input sequences are used. The queries $\boldsymbol{q}_i$ are created from one sequence by multiplication with $\boldsymbol{W}_q$, whereas the keys $\boldsymbol{k}_j$ and values $\boldsymbol{v}_j$ are created from another sequence by multiplications with $\boldsymbol{W}_k$ and $\boldsymbol{W}_v$, respectively. Some models such as the one proposed by Carion et al. (2020) even use three different input sequences.

For multi-head attention, the vectors $\boldsymbol{q}_i, \boldsymbol{k}_j$ and $\boldsymbol{v}_j$ are split into parts of equal size, for instance, $\boldsymbol{q}_i = \begin{bmatrix} \boldsymbol{q}_{i1}^T & \boldsymbol{q}_{i2}^T & \cdots & \boldsymbol{q}_{iN_H}^T \end{bmatrix}$, where $N_H$ indicates the number of heads. The dimensions of $\boldsymbol{q}_i, \boldsymbol{k}_j$ and $\boldsymbol{v}_j$, $D_q$ and $D_v$ must be integer-divisible by $N_H$.

Stacking the tokens into matrices $\boldsymbol{Q}^T = [\boldsymbol{q}_1 \ \cdots]$, $\boldsymbol{K}^T = [\boldsymbol{k}_1 \ \cdots]$ and $\boldsymbol{V}^T = [\boldsymbol{v}_1 \ \cdots]$, the attention mechanism proposed by Vaswani et al. (2017) is defined by

$$\hat{\boldsymbol{Q}} = \text{soft-argmax}\left(\eta \, \boldsymbol{Q}\boldsymbol{K}^T\right)\boldsymbol{V}, \tag{1}$$

where $\eta = D^{-1/2}$. The linear combinations of the rows of $\boldsymbol{V}$ yields the rows $\hat{\boldsymbol{q}}^T$ of matrix $\hat{\boldsymbol{Q}}^T$.

## 4   PROPOSED APPROACH: OPTIMIZING ATTENTION

The original transformer Vaswani et al. (2017) can be formulated as a message-passing algorithm, as shown by Veličković et al. (2018); Yun et al. (2019). We draw on this formulation and use a simplified intuition in the following: At each attention, the nodes in the *value* set $\{v_j\}$ send update messages $\boldsymbol{v}_j$ to the nodes of the *query* set $\{q_i\}$. The incoming messages into each node of the query set are weighted by the normalized scalar products $\text{soft-argmax}(\eta \boldsymbol{q}_i^T \boldsymbol{K}^T)$. These messages are sum-aggregated before the nodes in the query set are updated by an MLP. Normalization by soft-argmax maintains the original numerical ranges for the sum-aggregated update messages.

First, we formulate attention as a reconstruction problem, where we solve for the linear transformation that linearly combines the values so as to approximate the queries

$$\min_{\boldsymbol{x}_i} \left\| \boldsymbol{q}_i^T - \boldsymbol{x}_i^T \cdot \boldsymbol{V} \right\| \tag{2}$$

for a suitable norm, where $\boldsymbol{x}_i$ is an $N_v$-dimensional variable to be optimized. The scalars $\boldsymbol{x}_i$ control the linear combination and subsume the role of the attention coefficients $\text{soft-argmax}\left(\eta \, \boldsymbol{Q}\boldsymbol{K}^T\right)$ in Eq. equation 1.

In the case of cross-attention, the query and value tensors are the two inputs, respectively, without being transformed by the $\boldsymbol{W}_q$ and $\boldsymbol{W}_v$ weight matrices. By solving the optimization, we project the values to the subspace of the queries if the $l_2$-norm is being used; this resembles what the original cross-attention does. In self-attention, both query and value tensors are from the same input, so we require the $i$th element of $\boldsymbol{x}_i$ to be zero to avoid a trivial solution. In this way, the optimization reveals the correlation structure between every input element and the rest via this reconstruction.

Next, we impose the requirement that only few nodes in the value set should be allowed to send messages to a particular node $q_i$, which helps mitigate overfitting. Using an $l_1$ regularization, we arrive at the sparse reconstruction problem

$$\min_{\boldsymbol{x}_i} \left\| \boldsymbol{q}_i^T - \boldsymbol{x}_i^T \cdot \boldsymbol{V} \right\|_2^2 + \lambda \left\| \boldsymbol{x}_i \right\|_1, \tag{3}$$

where the scalar $\lambda$ controls the sparsity. We can solve Eq. equation 3 by means of the alternating direction method of multipliers (ADMM). Defining auxiliary variables $\boldsymbol{z}$, $\boldsymbol{\mu}$ and a scalar $\rho$, the ADMM can be optimized by iterating the following steps

$$\boldsymbol{x}^{(k+1)} = \left(\boldsymbol{V}\boldsymbol{V}^T + \rho\boldsymbol{I}\right)^{-1} \left(\boldsymbol{V}\boldsymbol{q}_i + \rho\left(\boldsymbol{x}^{(k)} - \boldsymbol{z}^{(k)}\right)\right) \tag{4a}$$

$$\boldsymbol{z}^{(k+1)} = \tau_{\lambda/\rho}\left(\boldsymbol{x}^{(k)} - \boldsymbol{\mu}^{(k)}\right) \tag{4b}$$

$$\boldsymbol{\mu}^{(k+1)} = \boldsymbol{\mu}^{(k)} + \boldsymbol{x}^{(k+1)} - \boldsymbol{z}^{(k+1)} \tag{4c}$$

where $\tau_a(\cdot)$ in Eq. equation 4b denotes the proximal operator, the superscript $(k)$ the iteration number and $\boldsymbol{I}$ the identity matrix. Since the token $\boldsymbol{q}_i$ is contained in one of the rows of $\boldsymbol{V}$ in case of self-attention, we zero-out the corresponding entry of $\boldsymbol{x}_i$ at each iteration. While the original convergence guarantee is not longer applicable, we notice that clamping the entries of $\boldsymbol{x}_i$ prevents divergence.

**Efficient Updates.** Naïvely inverting $\boldsymbol{V}\boldsymbol{V}^T + \rho\boldsymbol{I}$ in Eq. equation 4a is computationally expensive, since it is quadratic in the number of tokens in the value set and can be very large. Yet, we notice that $\boldsymbol{V}$ is extremely narrow; in fact, we usually have $N_v \gg D_v/N_H$. This implies that the eigenvalue decomposition $\boldsymbol{V}^T\boldsymbol{V} = \boldsymbol{X}\boldsymbol{D}\boldsymbol{X}^T$ can be efficiently computed. Since the non-zero eigenvalues of $\boldsymbol{V}\boldsymbol{V}^T$ equal those of $\boldsymbol{V}^T\boldsymbol{V}$, we only need the left singular vectors $\boldsymbol{Y} \approx \boldsymbol{V}\boldsymbol{X}\boldsymbol{D}^{-1/2}$ of $\boldsymbol{V} = \boldsymbol{Y}\boldsymbol{S}\boldsymbol{X}^{T}$[1] since they equal the eigenvectors of $\boldsymbol{V}\boldsymbol{V}^T$ corresponding to the non-zero eigenvalues. Thereby, we can efficiently compute the updates

$$\boldsymbol{x}^{(k+1)} = \boldsymbol{Y}\boldsymbol{S}^{-1}\left(\boldsymbol{Y}^T\left(\boldsymbol{V}\boldsymbol{q}_i + \rho\left(\boldsymbol{x}^{(k)} - \boldsymbol{z}^{(k)}\right)\right)\right) \tag{5}$$

without ever having to allocate memory for the large inverse. We augment matrix $\boldsymbol{S}$ by adding $\rho$ to the diagonal of $\boldsymbol{S} = \boldsymbol{D}^{1/2}$. To avoid exploding gradients during backpropagation, we add $\boldsymbol{r} \cdot \boldsymbol{I}$ to the diagonal of $\boldsymbol{V}^T\boldsymbol{V}$ where the entries of $\boldsymbol{r}$ are drawn from a uniform distribution $\mathcal{U}(0, \sigma)$ with $\sigma$ being small.

We notice that after some training, the matrix of values tokens $\boldsymbol{V}$ degenerates, i.e., many of the eigenvalues of $\boldsymbol{V}^T\boldsymbol{V}$ become very small whereby the minimal eigenvalue gap shrinks. This necessitates more iterations, thus slowing down forward and backward passes through the network and increasing memory demand. This problem does not appear in the original formulation of the ADMM due to the term $+\rho\boldsymbol{I}$ in Eq equation 4a which has the same effect as a Tikhonov-regularizer, i.e., it prevents rank-deficiency. Furthermore, the ADMM might not reach a reasonable solution if the query $\boldsymbol{q}_i$ is far from the range space $\text{span}(\boldsymbol{V}^T\boldsymbol{V})$.

**Random Token as Regularization.** Instead of reverting to the slow and memory-intensive formulation in Eq. equation 4a, we propose to create a set of tokens whose entries are drawn from a normal distribution with mean zero and unit covariance. These random tokens are then appended to $\boldsymbol{V}$. This has two consequences. First, it increases the eigenvalues of $\boldsymbol{V}^T\boldsymbol{V}$ thereby acting similarly as a Tikhonov-regularizer. Secondly, it endows the left singular vectors $\boldsymbol{Y}$ with components which span parts of the nullspace of $\text{span}(\boldsymbol{V}^T\boldsymbol{V})$, hence the distance between between query and range space reduces. In other words, the ADMM can better regress components of $\boldsymbol{q}_i$ in the kernel space of the non-augmented value matrix. We noticed that this reduces overfitting. The idea of using additional tokens is similar to the idea proposed by Darcet et al. (2024) except that the tokens used here are not trainable. Furthermore, they serve a specific purpose interpretable in terms of linear algebra.

Lastly, to further increase sparsity but avoid the computationally expensive soft-argmax operator, we map the estimated $\boldsymbol{x}_i$ by

$$\boldsymbol{x}'_i = \nu(\alpha(\boldsymbol{x}_i)), \tag{6}$$

where $\alpha(\cdot)$ be the function that raises each entry of $\boldsymbol{x}_i$ to its fifth power, and $\nu$ be a normalization function that, for instance, divides by the sum of the absolute values of $\boldsymbol{x}_i$. We observed that this reduces overfitting.

---

[1]Since it is standard to denote the value matrix by $\boldsymbol{V}$, we are using $\boldsymbol{V} = \boldsymbol{Y}\boldsymbol{S}\boldsymbol{X}^T$ for the singular value decomposition.

| | Model | AP | AP50 | AP75 | $AP_S$ | $AP_M$ | $AP_L$ |
|---|---|---|---|---|---|---|---|
| | DETR (Carion et al., 2020) | 40.6 | 61.6 | - | 19.9 | 44.3 | 60.2 |
| b-lines | enc SA wo weights | 37.7 | 59.0 | 39.1 | 17.0 | 41.3 | 55.7 |
| | dec SA wo weights | 38.5 | 58.0 | 40.3 | 17.5 | 41.3 | 58.3 |
| | dec CA wo weights | 36.0 | 57.7 | 37.2 | 15.1 | 38.4 | 55.6 |
| prop | prop encoder SA | 36.7 | 58.3 | 38.2 | 15.8 | 40.2 | 56.0 |
| | prop decoder SA | 38.5 | 57.9 | 40.1 | 18.8 | 41.5 | 57.4 |
| | prop decoder CA | 37.7 | 59.4 | 39.0 | 16.5 | 40.4 | 57.4 |

Table 1: Comparison between the three baseline models *baseline enc SA without weights*, *baseline dec SA without weights* and *baseline dec CA without weights* with the proposed algorithm (bottom three rows). The bottom row must be compared with *baseline dec CA without weights*, the second row from the bottom needs to be compared with *baseline dec CA without weights*, and the third row with *baseline enc SA without weights*. It can be seen that the proposed algorithm achieves superior results.

## 5 EXPERIMENTS

We evaluate the impact of the proposed optimization-based attention by comparing with transformer models that use both self- and cross-attention. The ADMM parameters are set to $\lambda = 0.1$, $\rho = 10$ and the number of iterations is taken to be 5 for all experiments.

We compare with the model (*DETR*) proposed by Carion et al. (2020) which comprises six transformer layers with self-attention, followed by another six layers with both self- and cross-attention. Specifically, we compare against several baselines that omit weight matrices in their attention mechanism, since our proposed approach also does not require weight matrices. We follow the procedure in (Carion et al., 2020) and perform experiments on the COCO 2017 detection dataset (Lin et al., 2014). As backbone, we use a ResNet50. We perform all experiments using 4 GPUs and a batch size of 8. We use the original learning rate, weight decay and learning rate scheduling.

### 5.1 COMPARISON WITH BASELINES

Table 1 shows results of the proposed method with several baselines. On the top are the results provided in the original *DETR* paper. The row below shows results of baselines which we trained for 300 epochs on 4 GPUs. The following 3 rows correspond to standard DETR versions that do not use weight projections in the self-attention of the encoder (*baseline enc SA wo weights*), the self-attention of the decoder (*baseline dec SA wo weights*) and the cross-attention of the decoder (*baseline dec CA wo weights*). The proposed attention model was evaluated with replacing the self-attention of the encoder (*prop encoder SA*), the self-attention of the decoder (*prop decoder SA*) and the cross-attention of the decoder (*prop decoder SA*). As such, we compare the first, second, and third variants of our proposed models with the first, second, and third baselines, respectively.

We see that our proposed approach performs similarly or better when comparing to the corresponding baselines. This shows that it is indeed possible to replace the standard attention mechanism with our proposed optimization-based approach while maintaining model performance. At the same time, our proposed approach does not require softmax and is by design parallelizable. When comparing with the original *DETR*, our models have lower detection scores since they use significantly less parameters (no weight matrices); for each attention layer in the encoder and decoder, we require 25% fewer parameters if one attention mechanism per layer is replaced or even 33% if both are replaced in the decoder.

### 5.2 ABLATIONS

In the ablation, we first analyze how the different cross-attention variants perform. We study the evolution from a plain $l_2$-regularized model ($l_2$-*optim*) in the first row of table 2 to the different variants of the efficient ADMM: the plain efficient ADMM indicated by (a) in the first column of table 2, the efficient ADMM endowed with 100 random tokens (b), with 500 random tokens and dropout augmentation (c), with 500 random tokens, dropout and sparsity inducing activation (SIA) in equation 6 (d), and like (d) but with soft path-dropout on the residual path (e). Here, we multiply the residual path around the attention with a random number $\tau \sim \mathcal{U}(0, 1)$ with probability $p = 0.5$.

| Model | $\Delta$ | AP | AP50 | AP75 | $AP_S$ | $AP_M$ | $AP_L$ |
|---|---|---|---|---|---|---|---|
| $l_2$-optim (CA) | 3.86 | 28.0 | 49.3 | 27.4 | 7.8 | 28.4 | 48.8 |
| efficient ADMM (CA) (a) | 3.53 | 30.8 | 51.8 | 30.5 | 10.1 | 32.5 | 51.5 |
| efficient ADMM (CA) (b) | 3.48 | 33.5 | 55.0 | 33.4 | 12.0 | 36.1 | 54.5 |
| efficient ADMM (CA) (c) | 3.52 | 33.4 | 54.5 | 33.5 | 12.0 | 35.6 | 54.3 |
| efficient ADMM (CA) (d) | 2.10 | 36.2 | 57.6 | 37.1 | 15.5 | 39.8 | 53.9 |
| efficient ADMM (CA) (e) | 1.78 | 37.7 | 59.4 | 39.0 | 16.5 | 40.4 | 57.4 |
| efficient ADMM (CA) (f) | 1.69 | 37.3 | 58.7 | 38.9 | 15.4 | 40.5 | 56.6 |
| efficient ADMM (CA) (g) | 1.73 | 37.1 | 57.9 | 38.5 | 15.4 | 39.7 | 57.1 |

Table 2: Ablation study: Comparison between different version of the algorithm when the cross-attention (CA) of the decoder is replaced. The second column ($\Delta$) indicates the average difference between test and training loss over the past 5 epochs. (a) the proposed efficient ADMM; (b) with 100 random tokens; (c) with 500 random tokens and dropout; (d) 500 random tokens, dropout and SIA; (e) 1000 random tokens, dropout and SIA; (f) like d with soft path dropout; (g) like d with hard path dropout.

| Model | $\Delta$ | AP | AP50 | AP75 | $AP_S$ | $AP_M$ | $AP_L$ |
|---|---|---|---|---|---|---|---|
| vanilla ADMM (SA) | 1.72 | 38.0 | 57.3 | 39.8 | 17.3 | 41.2 | 57.2 |
| efficient ADMM (SA) (a) | 2.02 | 38.5 | 57.9 | 40.1 | 18.8 | 41.5 | 57.4 |
| efficient ADMM (SA) (b) | 1.89 | 37.4 | 57.0 | 38.8 | 16.8 | 40.0 | 56.7 |

Table 3: Ablation study: Comparison between different version of the algorithm when the self-attention (SA) of the decoder is replaced. The second column ($\Delta$) indicates the average difference between test and training loss over the past 5 epochs. (a) the proposed efficient ADMM; (b) with 500 random tokens, dropout and SIA.

Finally, the last row (f) is like (d) but with hard path-dropout. Versions (a), (b), (c) and (g) were trained for 300 epochs, the other were terminated earlier at around 250 epochs.

These improvements are motivated by our observation that the network easily overfits. The gap between training and test losses ($\Delta$), averaged over the last 5 epochs, is given in the second column of table 2. It can be seen that all the precision metrics improve while the gap decreases from $>$ 3 to about 1.7. Most important improvements were the introduction of random tokens, sparsity inducing activation as well as a soft dropout on the residual path around the attention mechanism. We generally observe that convergence speed decrease for the more constrained variants, so using more epochs might be reasonable.

Similarly, we compare several variants for self-attention in the decoder in table 3. The first row corresponds to a vanilla ADMM while the second and third rows correspond to the plain efficient ADMM and the efficient ADMM with 500 random tokens and sparsity inducing activation, all without path dropout. Here, the precision metrics improve less over the computationally more expensive vanilla ADMM, probably due to the smaller gap between training and test losses already present in the vanilla ADMM case.

## 5.3 QUALITATIVE RESULTS

**Encoder, Self-Attention** We first show qualitative examples of the proposed algorithm used to replace the self-attention in the encoder. Figure 1 shows two examples with marked query points (left ear on top, below left eye) on the left and the corresponding attention heads on the right. Since the employed algorithm permits negative mixing coefficients, color differences indicate concentrated attention. Interestingly, with taking the left ear as query point, the attention focuses on the outlines of that ear in several heads while other heads remain inactive. Similarly for the left eye as query point; several heads focus on the eyes and even the nose, while several heads do nothing. There are more examples in the supplementary.

**Decoder, Cross-Attention** We further provide qualitative examples of the results of the proposed algorithm when the cross-attention in the decoder is replaced. The image from the validation set of MSCOCO shows three teddy bears which are indeed classified. Conversely, the corresponding baseline algorithm does not succeed. We select the head with the strongest classification score and show the attention maps of each head of the last layer in Fig. 2. It can be seen that the attention is focused on few parts of the image.

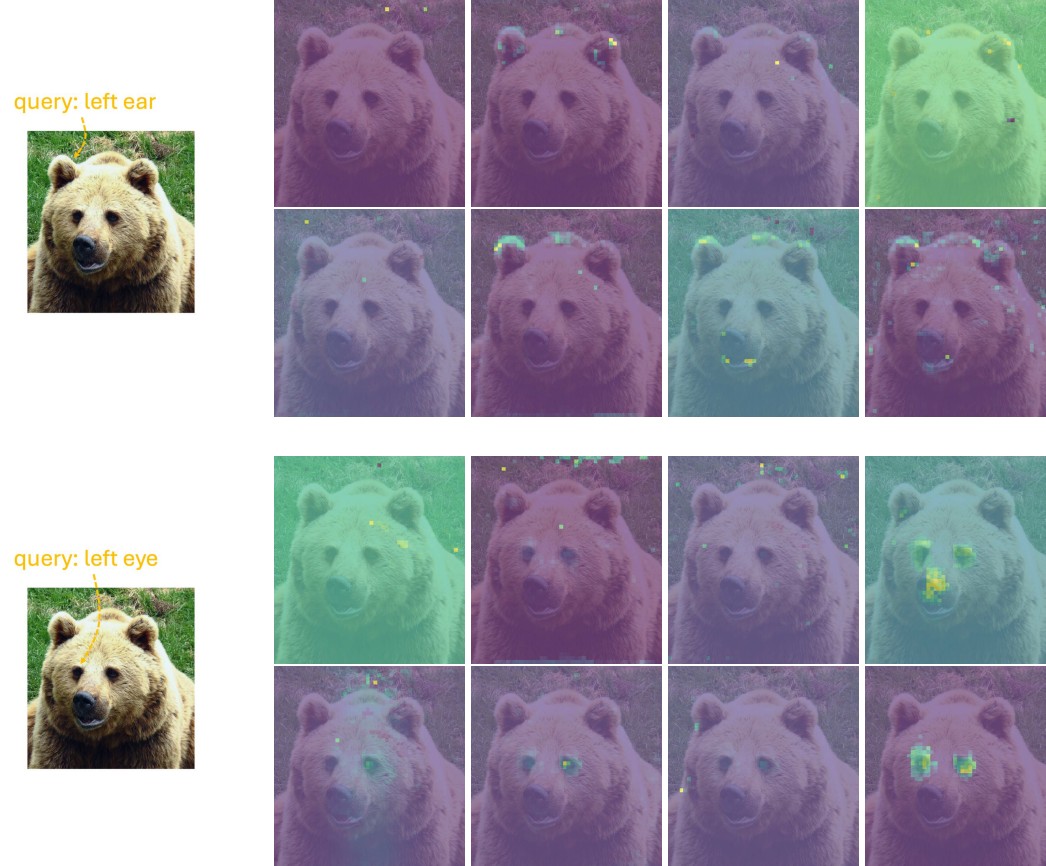

Figure 1: Two different examples of semantically meaningful attention focusing to particular queries at the last layer of encoder. The two images on the left indicate the two query points (left ear on top, left eye below). The two $2 \times 8$ sets of images on the right side show the images with the attention maps overlayed. Each each corresponds to one particular head. Please see the text for more explanation. It can be seen that some heads remain inactive while others focus on reasonable parts of the image. The image is from the validation set of MSCOCO.

The attention maps to the token with the second highest score are shown in Fig. 3. It can be seen that attention scores with large magnitude of several heads probe the outline of the shape. Lastly, we show the eight attention maps of the token with third highest classification score in Fig. 4. It can be seen that the attention is focused on the shape in the middle which is reasonable since the other two tokens are already focused on the outer two shapes.

For a different image, the attention maps of the token with the highest classification score are shown in Fig. 6. On the left is the original image, the maps corresponding to the eight heads are on the right. All heads concentrate on the *pot plant*.

Further examples are shown in the supplementary.

## 6 DISCUSSION

This work addresses the attention mechanism in transformers. It allows the network to compare tokens with each other. Standard transformers require that tokens are multiplied by weight matrices before and after the comparison. In particular on mobile devices, these weights can be so large that they do not fit into the cache memory of the compute units. This requires that subsets of tokens are multiplied with pieces of the matrices which in turn causes the same data to be transferred multiple

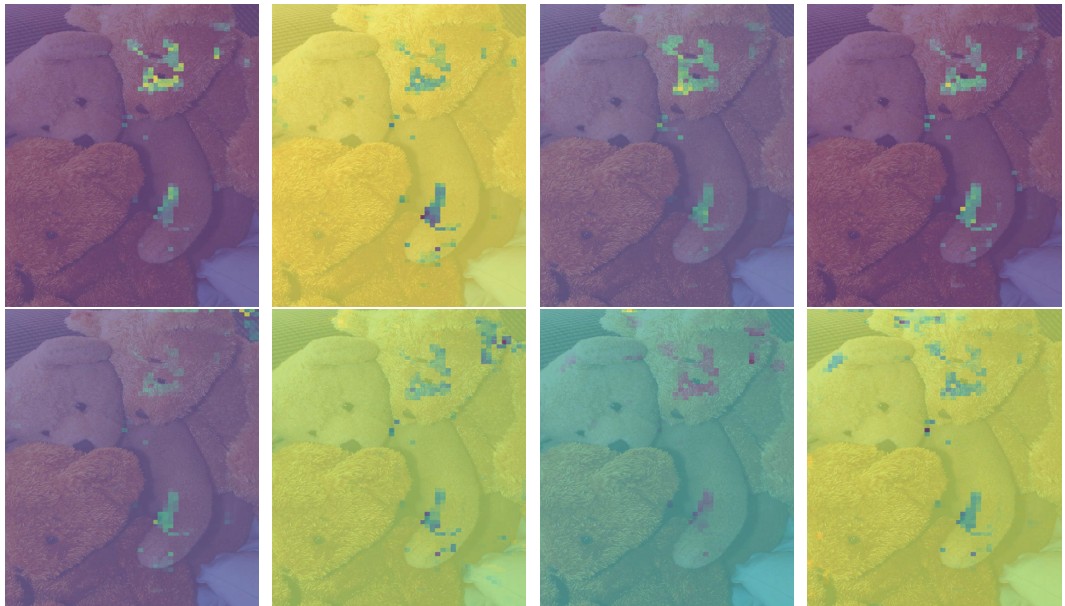

Figure 2: Attention maps of the last layer of the decoder overlayed onto the source image when the proposed algorithm is used for cross-attention. The maps stem from the token with the strongest classification score. The classification is correct. Indicative are color differences against the blue or yellow background because the proposed algorithm permits negative attention scores. The image is from the validation set of MSCOCO.

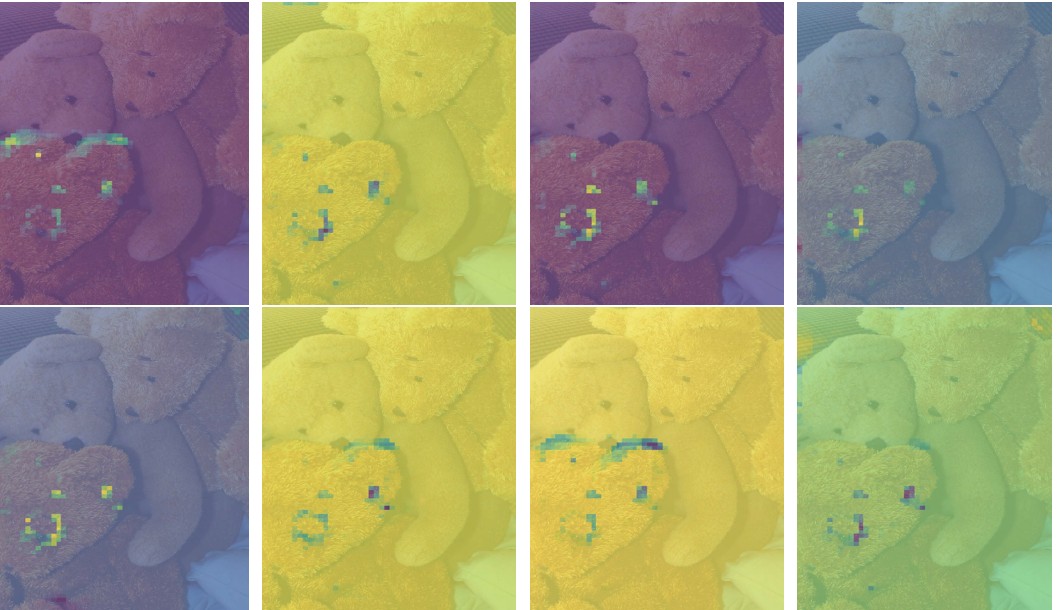

Figure 3: Attention maps of the last layer of the decoder overlayed onto the source image when the proposed algorithm is used for cross-attention. The maps stem from the token with the second strongest score. The classification is correct. The image is from the validation set of MSCOCO.

times between memory and cache. In particular if there is limited bandwidth, this operation can slow down the entire network.

While other works compress the data or propose how to minimize data transfers between the memories, we propose to eliminate the matrices from the attention algorithms. We show that atten-

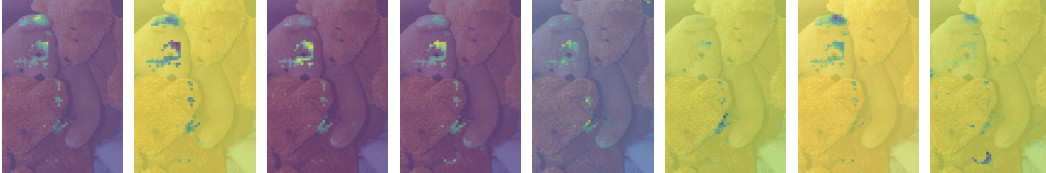

Figure 4: Attention maps of the token with third strongest (correct) classification score (cf. Figs. 2 and 3). It can be seen that the attention is focused on the shape in the middle. The image is from the validation set of MSCOCO.

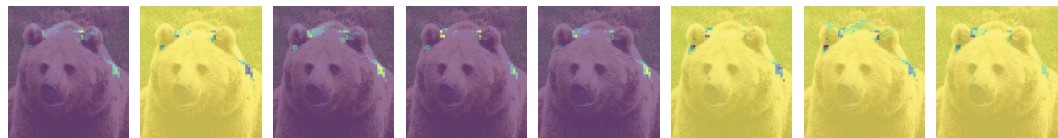

Figure 5: Attention maps of the token with highest classification score for the *bear* image. It can be seen that the attention is probes the shape of the animal. The image is from the validation set of MSCOCO.

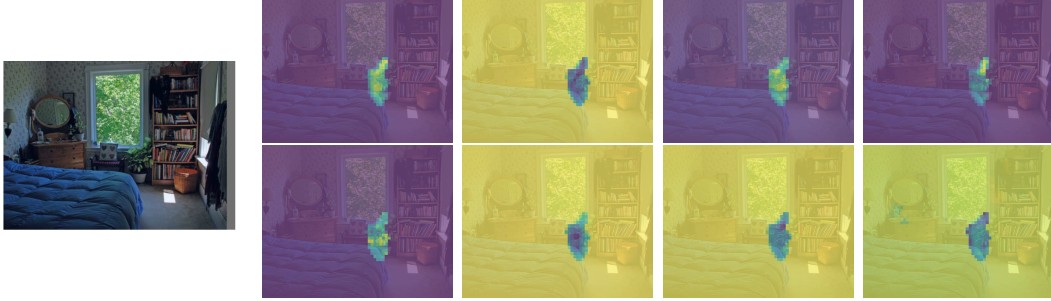

Figure 6: Decoder cross-attention: Shown is where the token with the highest classification score attends to. Left: original image. Right: eight attention heads. The token correctly classifies a *pot plant*. The image is from the validation set of MSCOCO.

tion can be performed by solving optimization problems, thus the optimal attention coefficients are determined in a data-driven way. To increase the speed of those operations and allow for auto-differentiation, we propose several techniques to reduce the required memory. The experimental evaluation shows that our algorithms compare favorably with the baselines and that the attention maps can be semantically reasonably. We hope that this work provides a novel perspective on the mechanism of attention.

## 7    REPRODUCIBILITY

We integrated our algorithm in the excellent repository provided by the authors of DETR (Carion et al., 2020). After reshaping the token tensors into the heads, Eq. 4 needs to be iterated. Afterwards, the attention scores need to be multiplied with the mask tensor to account for padded images. Lastly, the scores are multiplied with the value tensor. For our efficient variants, Eq. 5 can be used. If random tokens are to be used, they need to be appended to the value tensor.

## 8    ETHICS STATEMENT

This research was undertaken with having best practices for scientific research in mind. No experiments with human were performed during this project. We used standard, publicly available data to allow for comparison with others in the field. We made sure not include any image of persons into this paper so as not to violate any privacy. We adhere to strict protocols for data collection and storage. The paper was third-party inspected for possible violations of these policies. We complied with all relevant legal and ethical standards throughout the research process.

We recognize that unforeseen biases can result from the limited MSCOCO dataset, or even intentional misuses. We thus understand this research project only as inspiration to others in the field. Before deploying the developed algorithm to real-world applications, strict tests with much more data have to be undertaken to ensure that unintentional biases are not present and misuses may not occur.

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

## A  QUALITATIVE RESULTS BASELINE

We show several qualitative examples of attention maps using the vanilla baseline model with a checkpoint provided by Carion et al. (2020).

### A.1  ENCODER SELF-ATTENTION

Here, we show examples of attention maps at the last layer of the encoder. The image is from the test set of MSCOCO. For the attention maps in Fig. 7, we chose a point at the left ear. It can be seen that one of the heads indeed focuses at the left ear. For the maps in Fig. 8 we chose a point at the nose of the bear. Here, no head focuses at the nose.

### A.2  DECODER CROSS-ATTENTION

Figs. 9 and 10 show examples to which image areas query tokens of the decoder focus to. Some tokens appear to probe semantically meaningful parts of the image whereas other tokens seem to remain inactive.

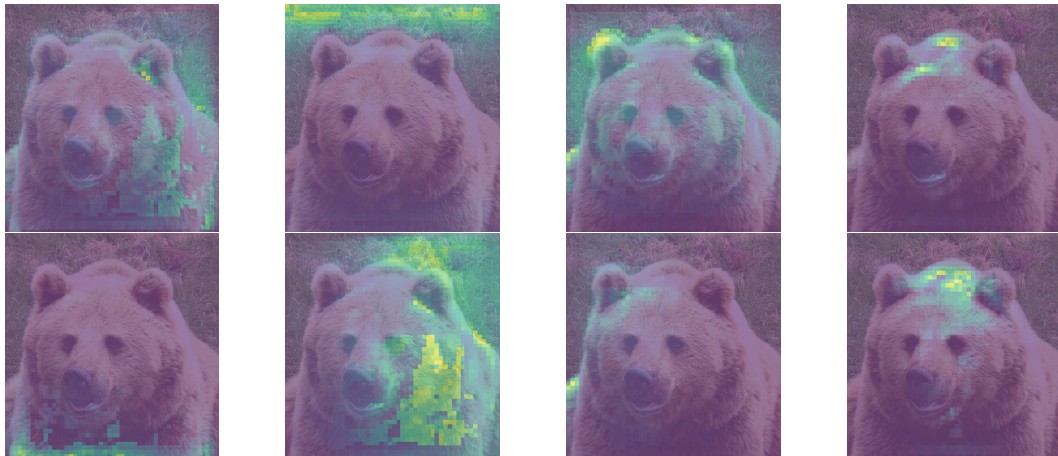

Figure 7: Qualitative examples of the image that the encoder attends to when the left ear is queried. It can be seen that one head strongly focuses on the nose yet the other heads attend to other image areas. The image is from the validation set of MSCOCO.

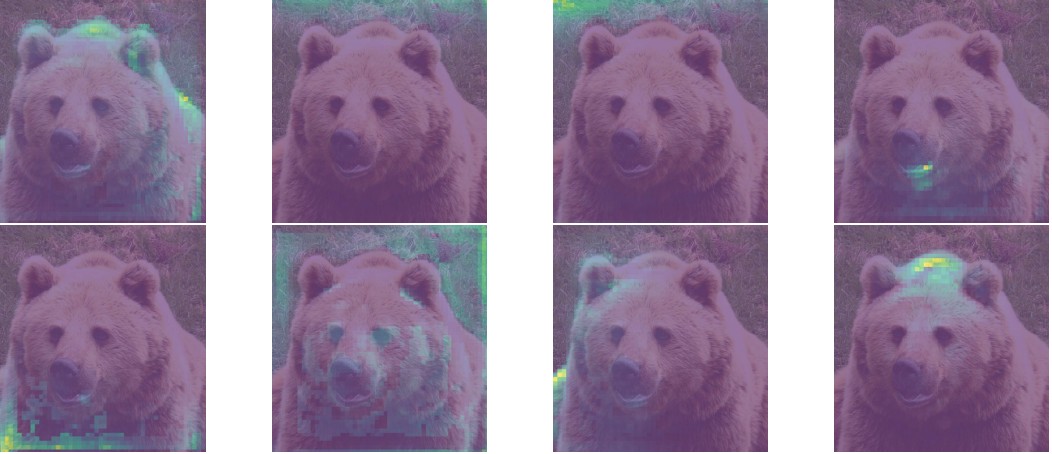

Figure 8: Qualitative examples of the image that the encoder attends to when a point at the nose is queried. It can be seen that the attention does not focus on the correct spot for any head. The image is from the validation set of MSCOCO.

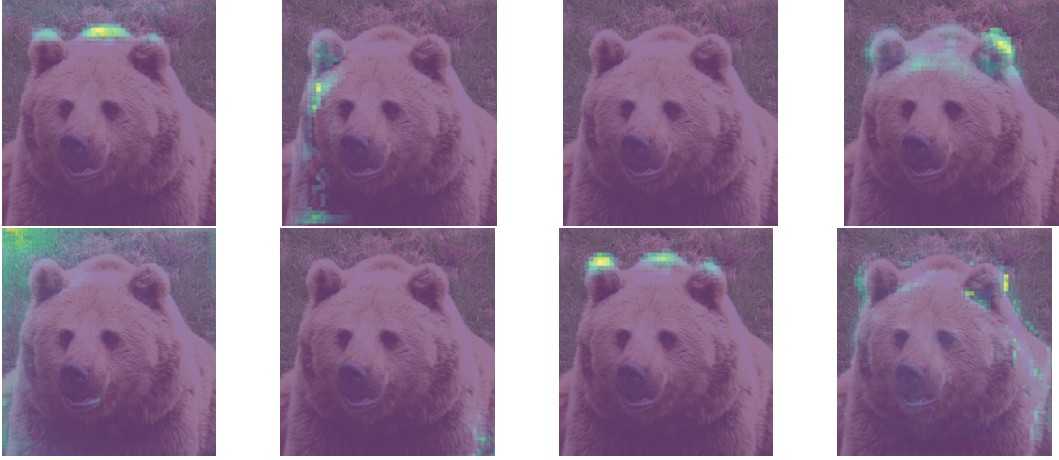

Figure 9: Qualitative examples of the image that one of the query tokens of the decoder attends to. It appears as if the token mainly probes both ears and the head. The image is from the validation set of MSCOCO.

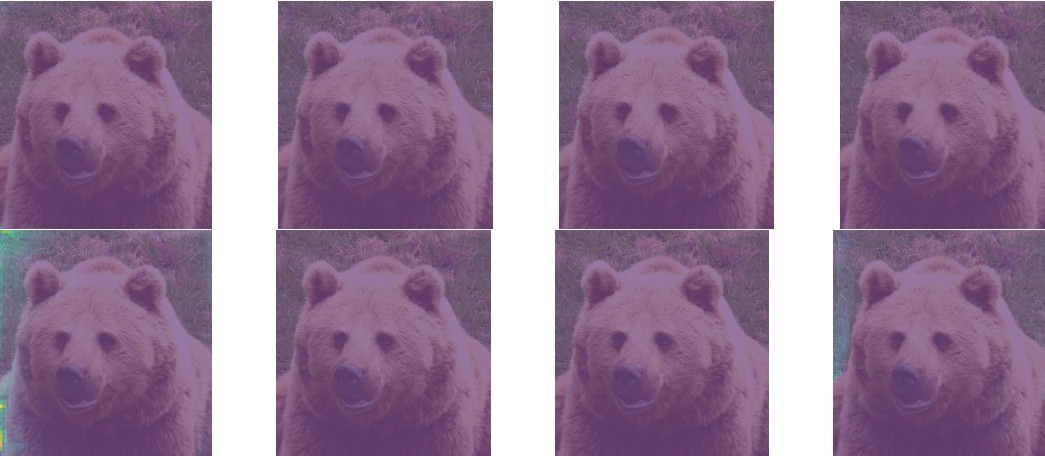

Figure 10: Qualitative examples of the image that one of the query tokens of the decoder attends to. Apparently, there is little focus on any part of the image. Indeed, we found that many tokens remain inactive. The image is from the validation set of MSCOCO.

## B   QUALITATIVE RESULTS PROPOSED ALGORITHM

We show more examples of attention maps estimated by the proposed algorithm.

### B.1   ENCODER, SELF-ATTENTION

In this section, we show more examples of the attention maps stemming from replacing the self-attentions in the encoder. Since the attention map yields scores between each (query) token and all others, it is quadratic. For a visual inspection, we therefore need to select a query token for which we can then visual the 2D attention map.

Figure 11 shows more examples of attention maps when the query point is the nose of the bear.

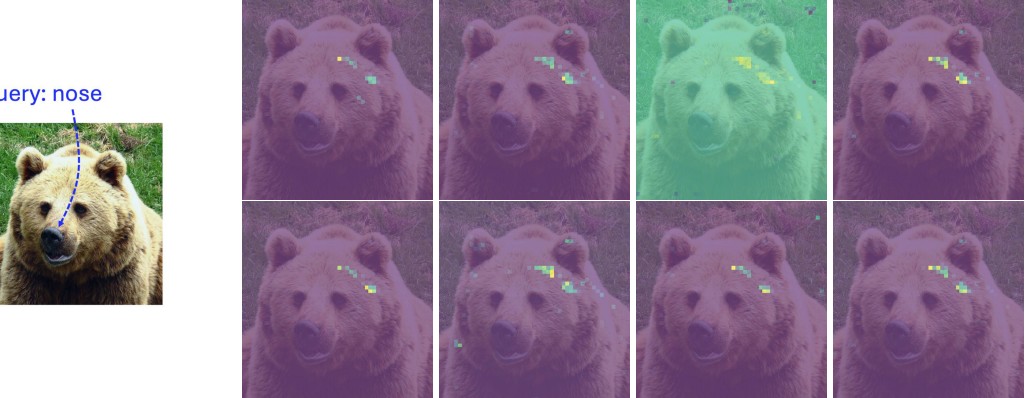

Figure 11: Examples of the attention map caused by the query point shown in the left image. The attention appears to be sparse. The image is from the test set of MSCOCO.

### B.2   DECODER, CROSS-ATTENTION

In this section, we show more examples of attention maps arising from replacing the cross-attentions in the decoder with the proposed algorithm.

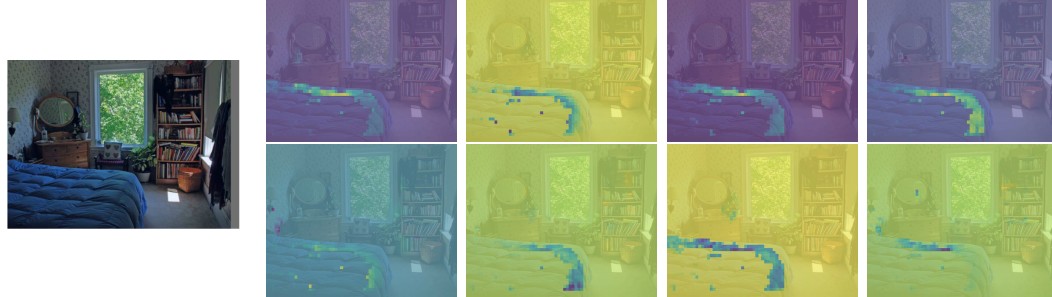

Figure 12: Decoder cross-attention: Shown is where the token with the second highest classification score attends to. Left: original image. Right: eight attention heads. The image is from the validation set of MSCOCO.

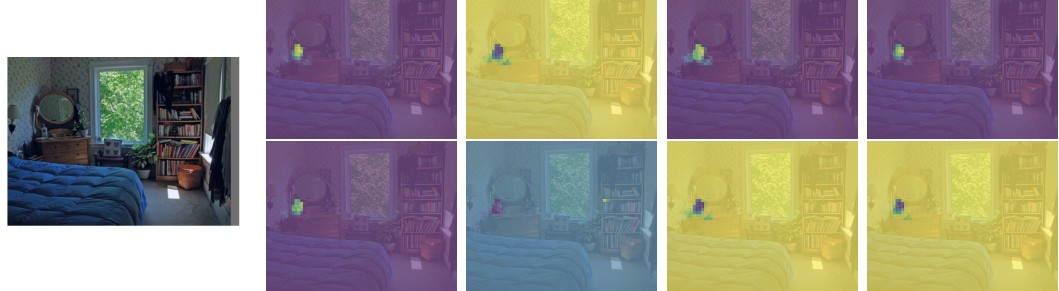

Figure 13: Decoder cross-attention: Shown is where the token with the third highest classification score attends to. Left: original image. Right: eight attention heads. The image is from the validation set of MSCOCO.

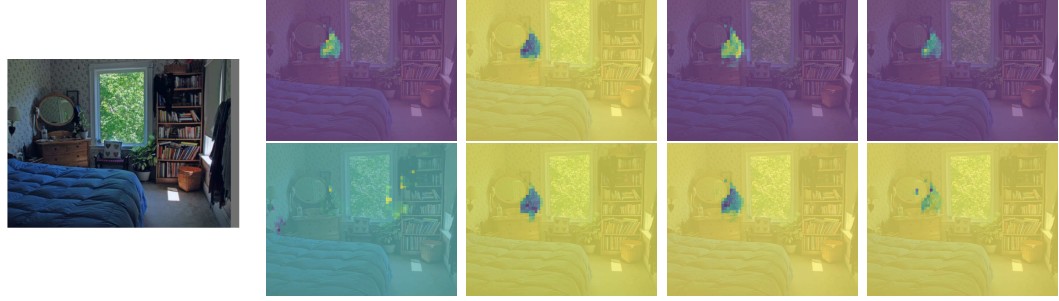

Figure 14: Decoder cross-attention: Shown is where the token with the fourth highest classification score attends to. Left: original image. Right: eight attention heads. The image is from the validation set of MSCOCO.

