# OpenReview forum: "Optimizing Attention"
_ICLR.cc/2025/Conference — ICLR 2025 Conference Withdrawn Submission_

### Official Review · Reviewer_2gw5 · 2024-10-15

**Soundness:** 1
**Presentation:** 1
**Contribution:** 2
**Rating:** 1
**Confidence:** 4

**Summary:**

The paper proposes a method of replacing the standard QKV attention module in transformers with a method that approximates attention without the need for explicit Q and K projection matrices. This is achieved by performing a fixed number of steps of an optimization problem that tries to find a sparse linear combination of values.

**Strengths:**

The key strength of the paper is in its novel approach to approximate attention. As mentioned in the introduction, there are attention speedup methods which fall into categories such as reformulation (e.g. linear attention [1]), attention matrix decomposition (e.g. Linformer [2], Performer [3]), and hardware aware algorithms (e.g. FlashAttention [4]). I haven't come across an approach that tries to meta-learn the attentional value update as a sparse linear combination of the inputs.


[1] [Transformers are RNNs: fast autoregressive transformers with linear attention](https://dl.acm.org/doi/abs/10.5555/3524938.3525416)
[2] [Linformer: Self-Attention with Linear Complexity](https://arxiv.org/abs/2006.04768)
[3] [Rethinking Attention with Performers](https://openreview.net/forum?id=Ua6zuk0WRH)
[4] [FlashAttention: Fast and Memory-Efficient Exact Attention with IO-Awareness](https://dl.acm.org/doi/10.5555/3600270.3601459)

**Weaknesses:**

There are severe methodological issues with this paper.


At a high level, if the motivation in the introduction for the work is the reduce the memory requirements for running a transformer by eliminating the Q and K matrices, in an effort to have less cache thrashing on edge devices (lines 032-035), then the experimental section needs to provide data on how the proposed intervention improves on this aspect. I would expect to see strong baselines against FlashAttention(1-3) in particular, as they're methods to run an arbitrary transformer without modification. To this effect, it appears as though only Table 1 is comparing against baseline methods.


Table 1 - What is immediately obvious is that the DETR baseline is superior at every average precision level. The key difference, I suppose is that it has Q and K projection matrices, which this paper wants to eliminate. Presumably, the next set of baselines are DETR, but with identity mappings instead of QK projections. I would have liked to see this choice motivated by latency/throughput analysis on the target device. The expectation would be that it's much faster (due to claimed reduction of cache requirement), but also noticeably worse (this table). Regardless, the only place where the proposed method seems to be consistently better is with decoder cross attention, as the two self-attention rows seem to be worse than the baseline. The discussion on lines 254-257 "This shows that it is indeed possible to replace the standard attention mechanism with our proposed optimization-based approach while maintaining model performance." is not consistent with the caption in Table 1 "It can be seen that the proposed algorithm achieves superior results." which makes it unclear what is being compared. Aside from conjecture, the paper doesn't demonstrate that it is faster, or even necessarily that it's theoretically faster, on device.

_"At the same time, our proposed approach does not require softmax and is by design parallelizable."_ - This would be an excellent opportunity to demonstrate how this actually affects latency/throughput on modern hardware.

Equation 6 - Given that $\nu$ is a reduction operation, are we comparing the computation expense of $\alpha(x) = x^5$ versus $e^x$ as in softmax?

General methodology issues:
* There are 4 usages of "We notice", but none of them seem to be accompanied by empirical or theoretical evidence.
* It would good to see comparisons with similar-in-class methods. Given DeTr's age (4 years), certainly there have been subsequent works. Do any of those also address edge device efficiency? If so, they need to make it into Table 1.

**Questions:**

I am unclear on how "In addition, the attention operation incurs quadratic complexity w.r.t. number of input tokens, both in terms of memory usage and computation." relates to equation 2, which still seems to suggest $O(n^2)$ scaling, as the update still seems to involve a combination of all values for each query.

Attention is defined using softmax, not soft-argmax. Is there a subtlety to the method that relies on soft-argmax, or is this a mistake in the paper? Presumably soft-argmax would be akin to gumbel-softmax sampling, which would be 1-hot fetching (indeed, this would be memory efficient, but I don't think it's the intent).

I think the paramount question I have is this though: Is this method in any way faster than FlashAttention? I need to see experimental evidence for this.

---

### Official Review · Reviewer_EG1S · 2024-10-30

**Soundness:** 2
**Presentation:** 3
**Contribution:** 2
**Rating:** 3
**Confidence:** 4

**Summary:**

The paper introduces a novel attention mechanism that eliminates trainable parameters. Instead of computing an attention matrix and applying the SoftMax operation, the authors propose optimizing a coefficient matrix by iteratively minimizing the distance between query and value pairs. Additionally, the paper introduces sparsification and random token techniques for regularization. The proposed method is evaluated by replacing various self-attention and cross-attention components in the DETR model on the COCO 2017 detection dataset.

**Strengths:**

1. The exploration of parameter-free attention is intriguing. If successful, it could significantly enhance models that rely on attention mechanisms.
2. The proposed attention mechanism is novel, to the best of my knowledge.
3. The explanation of the attention mechanism is clear and easy to follow.

**Weaknesses:**

1. The proposed method introduces new hyperparameters, such as the lambda in Equation 3 and the number of random tokens for regularization. These hyperparameters are not automatically optimized and may require tedious manual tuning.
2. The experimental setup has several issues:
   a. The choice to use DETR without weights as a baseline seems questionable. The paper critiques attention for its large matrices in mapping inputs to queries, keys, and values. Therefore, a comparison with the original DETR, including its performance, would be more appropriate than comparing it to variants without weights.
   b. The paper overlooks important efficiency metrics, such as latency, model size, and memory consumption.
   c. An experiment fully replacing both self-attention (SA) and cross-attention (CA) with the proposed attention mechanism is missing. This would better demonstrate the method's effectiveness in replacing traditional attention mechanisms.
   d. The paper should consider comparisons with additional baselines, such as simple pooling techniques, as explored in PoolFormer.
   e. Evaluating only on detection tasks is insufficient. Including experiments on other modalities and tasks, such as text processing, which depends more heavily on long-range context modeling, would strengthen the paper.

**Questions:**

Beyond the weaknesses mentioned above, the current title appears inappropriate. "Optimizing Attention" suggests the paper focuses on optimizing existing attention mechanisms, whereas it actually proposes a new attention method based on iterative optimization. A more precise title would better reflect the paper's contributions.

---

### Official Review · Reviewer_acUC · 2024-10-31

**Soundness:** 2
**Presentation:** 2
**Contribution:** 2
**Rating:** 3
**Confidence:** 3

**Summary:**

This paper presents a novel attention approach for transformer.
Specifically, it transforms the original problem into an optimization problem and provides some techiniques for efficient update.
Experimental results show some potenal for this kind of new attention approach.

**Strengths:**

1. The proposed optimization-based attention mechanism is novel and offers a fresh perspective on designing attention mechanisms without relying on large weight matrices or softmax operations.
2. The approach has the potential to reduce memory usage and parameter count.
3. The theoretical framework, especially the use of ADMM for sparse reconstruction, introduces new insights into the inner workings of attention mechanisms.

**Weaknesses:**

I have several concerns regarding the proposed approach:
1. Computational Complexity: The proposed method involves computationally expensive operations, such as singular value decomposition (SVD) and matrix inversion, which raises concerns about the scalability of the model for large-scale training. I would encourage the authors to provide a detailed complexity analysis comparing their approach to standard attention mechanisms, especially with regard to the SVD and matrix inversion operations. Additionally, runtime comparisons on larger datasets would help to clarify the implications of these operations for scalability.
2. Evaluation Limited to Computer Vision: The experimental evaluation focuses solely on computer vision datasets, raising concerns about the method's generalizability to other domains, such as natural language processing (NLP). Given the widespread use of transformers in NLP tasks, I recommend that the authors expand their evaluation to include popular NLP benchmarks such as machine translation (e.g., WMT), text classification (e.g., GLUE), or question answering (e.g., SQuAD). This would provide a more comprehensive assessment of the method’s versatility across different applications.
3. Manuscript Clarity and Structure: The paper would benefit from more thorough polish and a clearer exposition of certain sections. For instance, the explanation of the proposed optimization approach could be more detailed, particularly in how the random token augmentation and the efficient updates interact with the ADMM framework. Moreover, the experimental section could be expanded to discuss the results more comprehensively, including potential limitations and future work. Rather than suggesting a strict target page count, I recommend focusing on clarifying and expanding specific sections that currently feel underdeveloped or rushed.

Lastly, I would gently suggest that the authors aim for a manuscript of approximately 10 pages. The current version appears rushed and would benefit from more thorough polish.

**Questions:**

Please refer to the weakness part.

---

### Official Review · Reviewer_9CXh · 2024-11-01

**Soundness:** 2
**Presentation:** 3
**Contribution:** 2
**Rating:** 5
**Confidence:** 2

**Summary:**

The paper presents an efficient optimized version of attention operation to be used in edge devices. The idea is reformulate attention as an optimization problem. Replaces softmax with a different function that raises input to 5th power and then normalizes rather than using exponentiation.

**Strengths:**

The paper approaching efficient attention computation by redefining attention as an optimization problem and using clever methods to compute the approximate attention using efficient updates and random token regularization.

**Weaknesses:**

Line 142: The statement is not clear.  "These messages are sum-aggregated before the nodes in the query set are updated by an MLP". How did MLP come in the explanation of self-attention operation between query and value.
The results are only done for the tast of object detection. In otder to better evaluate the performance of this new optimization stretegy, the results of other tasks such as image classification, segmentation etc should be shown.
In Table 1 most of the comparative results with baselines shows that this method is inferior to baselines.
Even though the proposed method talks about the efficiency due to not storing attention matrices, it does not shows any efficiency metric such as latency, FLOPs, parameter count, peak memory etc.
The results shown in qualitative part is inconclusive.

**Questions:**

It is not clear why the linear combination of values should approximate the query. Can you explain in detail?

---

### Note · Authors · 2024-11-22

**Comment:**

We thank all the reviewers for their time and effort and their helpful comments about this paper.

**Withdrawal Confirmation:**

I have read and agree with the venue's withdrawal policy on behalf of myself and my co-authors.